



# Sulfuric acid-amine nucleation in urban Beijing

Runlong Cai[1,2,3], Chao Yan[1,3], Dongsen Yang[4], Rujing Yin[2], Yiqun Lu[5], Chenjuan Deng[2], Yueyun Fu[2], Jiaxin Ruan[4], Xiaoxiao Li[2], Jenni Kontkanen[3], Qiang Zhang[2], Juha Kangasluoma[1,3], Yan Ma[4], Jiming Hao[2], Douglas R. Worsnop[6], Federico Bianchi[3], Pauli Paasonen[3], Veli-Matti Kerminen[3], Yongchun Liu[1], Lin Wang[5], Jun Zheng[4], Markku Kulmala[1,3], and Jingkun Jiang[2]

[1]Aerosol and Haze Laboratory, Beijing Advanced Innovation Center for Soft Matter Science and Engineering, Beijing University of Chemical Technology, Beijing, 100029, China
[2]State Key Joint Laboratory of Environment Simulation and Pollution Control, School of Environment, Tsinghua University, Beijing, 100084, China
[3]Institute for Atmospheric and Earth System Research / Physics, Faculty of Science, University of Helsinki, Helsinki, 00014, Finland
[4]Collaborative Innovation Center of Atmospheric Environment and Equipment Technology, Nanjing University of Information Science and Technology, Nanjing, 210044, China
[5]Shanghai Key Laboratory of Atmospheric Particle Pollution and Prevention (LAP[3]), Department of Environmental Science and Engineering, Fudan University, Shanghai, 200433, China
[6]Aerodyne Research Inc., Billerica, Massachusetts, 01821, USA

*Correspondence to*: Jingkun Jiang (jiangjk@tsinghua.edu.cn)

**Abstract.** New particle formation (NPF) is one of the major sources of atmospheric ultrafine particles. Due to the high aerosol and trace gas concentrations, the mechanism and governing factors for NPF in the polluted atmospheric boundary layer may be quite different from those in clean environments, which is however less understood. Herein, based on long-term atmospheric measurements from January 2018 to March 2019 in Beijing, the nucleation mechanism and the influences of $H_2SO_4$ concentration, amine concentrations, and aerosol concentration on NPF are quantified. The collision of $H_2SO_4$-amine clusters is found to be the dominating mechanism to initialize NPF in urban Beijing. The coagulation scavenging due to the high aerosol concentration is a governing factor as it limits the concentration of $H_2SO_4$ amine clusters and new particle formation rates. Besides, the effective amine concentration is another limiting factor in Beijing because amine is sometimes insufficient for nucleation at the kinetic limit. Based on the synergistic effects of these factors on $H_2SO_4$-amine nucleation, governing factors for $H_2SO_4$-amine nucleation for different conditions are summarized.





## 1 Introduction

New particle formation (NPF) is a major source of ambient particles in terms of particle number concentration. During a typical NPF process, gaseous precursors form stable clusters via nucleation. Some of these clusters survive and grow into cloud condensation nuclei and hence have the potential to influence the global climate (Kuang et al., 2009; Gordon et al., 2017). There has been a considerable number of NPF studies in various atmospheric environments (Kulmala et al., 2004; Kerminen et al., 2018; Lee et al., 2019), but current knowledge on NPF in the polluted atmospheric boundary layer (e.g., the urban environment in megacities) is still limited. In the presence of a high aerosol concentration in the polluted environment, a considerable fraction of the newly formed clusters and particles are scavenged by coagulation within minutes and hence, NPF may be significantly suppressed (McMurry et al., 2005; Kuang et al., 2010). However, frequent NPF events with high formation rates have been reported in polluted environments (Wu et al., 2007; Iida et al., 2008; Wang et al., 2015; Cai and Jiang, 2017; Yao et al., 2018; Deng et al., 2020b). Such unique characteristics of NPF with a high aerosol concentration, moderate gaseous precursor (e.g., $H_2SO_4$) concentrations, and a high particle formation rate indicate a fast nucleation mechanism in these environments. Various nucleation mechanisms for the real atmosphere have been reported, such as $H_2SO_4$-amine nucleation (Chen et al., 2012; Yao et al., 2018), $H_2SO_4$-$NH_3$ nucleation (Chen et al., 2012; Jokinen et al., 2018), oxidized organics nucleation (Bianchi et al., 2016), and $HIO_3$ nucleation (Sipila et al., 2016). The second and third mechanisms are not efficient enough to explain the observed high particle formation rates in the polluted environment and the last mechanism may dominate NPF only in the coastal regions.

The clustering of $H_2SO_4$ and amines produces new particles at a high formation rate. Laboratory studies showed that some amines enhance $H_2SO_4$ nucleation more efficiently than $NH_3$ (Kirkby et al., 2011; Erupe et al., 2011; Yu et al., 2012; Jen et al., 2014; Dunne et al., 2016; Yu et al., 2018). Atmospheric measurements in Boulder (Zhao et al., 2011) and urban Atlanta (Chen et al., 2012) indicate the $H_2SO_4$-amine nucleation mechanism. The CLOUD (cosmics leaving outdoor droplets) chamber experiments and theoretical calculations based on quantum chemistry reported that in the presence of ~5 ppt dimethylamine (DMA, $(CH_3)_2NH$) as the stabilizing base, $H_2SO_4$ under a typical atmospheric concentration (~$10^6$ - $10^7$ molecules·$cm^{-3}$) nucleates at a rate approaching the kinetic collision limit where cluster evaporation is negligible (Almeida et al., 2013; Kürten et al., 2014; Kürten et al., 2018). Atmospheric measurements in urban Shanghai (Yao et al., 2018) provide supports for the view that in polluted megacities in China, $H_2SO_4$ initiates NPF and DMA is perhaps the dominating base to stabilize $H_2SO_4$ clusters. Elucidating the governing factors for atmospheric nucleation and their quantitative impacts on particle formation rate is a key to understanding the nucleation mechanism in the real atmosphere.

Previous laboratory experiments and atmospheric measurements provide quantitative understandings of $H_2SO_4$-DMA nucleation (Chen et al., 2012; Almeida et al., 2013; Jen et al., 2014; Kürten et al., 2014; Kürten et al., 2018). Due to the high aerosol concentration in the polluted atmospheric boundary layer, the loss rates of new particles during NPF in urban Beijing and Shanghai are usually ~10 times higher than the total loss rate in chambers. Under such a high aerosol concentration, coagulation scavenging has been found to govern the concentrations of new particles (Cai et al., 2017b; Deng et al., 2020a).



In addition, high DMA concentrations were used in the previous chamber experiments, e.g., up to 200 ppt in Jen et al. (2014) and up to 140 ppt in the CLOUD experiments (Almeida et al., 2013; Kürten et al., 2014; Kürten et al., 2018), whereas the
DMA concentration in urban Beijing was observed to be usually lower than 5 ppt in this study. As also pointed out in previous studies (Kürten et al., 2014; Yao et al., 2018), the characteristics and limiting factors of nucleation in the polluted atmosphere may be different from those in laboratory experiments, due to differences in the particle loss rate and precursor gas concentrations. As a result, the molecular scale understanding of $H_2SO_4$-DMA nucleation under laboratory conditions (Jen et al., 2014; Kürten et al., 2014; Kürten et al., 2018) may not be directly applicable for the real atmosphere with low
DMA concentrations (< 5 ppt) and high aerosol concentrations.

For a better understanding of the nucleation mechanism in the polluted atmosphere, long-term atmospheric measurements were conducted in urban Beijing from January 2018 to March 2019. Gaseous $H_2SO_4$ and cluster concentrations, amine concentrations, and particle size distributions ranging from 1 nm to 10 μm were measured. The formation mechanism and the factors governing the initial steps of NPF in the polluted environment are explored. A model
based on kinetic nucleation theory is shown to well predict the concentrations of $H_2SO_4$ dimer, trimer, and tetramer and formation rate of 1.4 nm particles in urban Beijing. The roles of coagulation scavenging and amine concentrations in $H_2SO_4$-amine nucleation are revealed and quantified.

## 2 Measurements

The atmospheric measurement was conducted in urban Beijing. The dataset for this study is from January 2018 to March
2019. The observation site is located at the campus of Beijing University of Chemical Technology (39°56′ N, 116°17′ E). The west 3$^{rd}$ ring road is ~500 m away from this observation site. State-of-the-art instruments were deployed to capture the whole NPF process from the very initial step of nucleation to particle growth. The aerosol size distributions ranging from 1 nm to 10 μm were measured using a diethylene glycol scanning mobility particle spectrometer (1-4.5 nm, Jiang et al., 2011; Cai et al., 2017a; Fu et al., 2019) and a particle size distribution system (3 nm - 10 μm, Liu et al., 2016). The neutral gaseous
$H_2SO_4$ molecule and cluster concentrations were measured using chemical ionization time-of-flight mass spectrometers (ToF-CIMS, Aerodyne Research, Inc., Jokinen et al., 2012). $H_2SO_4$ molecules and neutral clusters are charged using a nitrate chemical ionization source. A long ToF-CIMS and a high-resolution ToF-CIMS were used before and after September 2018, respectively. These two instruments were calibrated separately and compared during a short-term parallel measurement. The $H_2SO_4$ monomer concentrations reported by these two instruments agree with each other within a systematic relative
difference of 1.4±0.3 (Fig. S1). The amine molecule in a stable neutral $H_2SO_4$-amine cluster may detach during the detection ToF-CIMS. Hence, the concentrations of $H_2SO_4$ clusters containing the same number of $H_2SO_4$ molecules were summed up. $NH_3$ was not found in the detected neutral $H_2SO_4$ clusters. In additions to $H_2SO_4$, organics with low volatilities were measured using ToF-CIMS. Neutral amine and $NH_3$ concentrations were measured using a modified ToF-CIMS (Aerodyne Research, Inc., Zheng et al., 2015) since October 2018. The reagent ions to charge amines and $NH_3$ are $H_3O^+$ or its hydrated





clusters. The sampling line for this ToF-CIMS was ~1.5 m long and it was heated to ~60 °C. The temperature of the sampled air was < 40 °C. A sample flow rate of 6.1 L·min$^{-1}$ was used to reduce aerosol deposition onto the tube wall. Because the ToF-CIMS cannot separate isomers, the measured C$_2$-amine concentration is taken as DMA concentration. Ethylamine is thought to be less efficient as a stabilizing base for (H$_2$SO$_4$)$_1$(amine)$_1$ than DMA (Xie et al., 2017), thus, the measured effective DMA concentration for stabilizing H$_2$SO$_4$ clusters might be overestimated. Similarly, the measured C$_3$-amine

concentration is taken as trimethylamine (TMA) concentration. The naturally charged negative clusters were measured using an atmospheric pressure interface time-of-flight mass spectrometer (APi-ToF-MS, Aerodyne Research, Inc., Junninen et al., 2010). Ambient temperature, pressure, and relative humidity were measured using a weather station (AWS310, Vaisala Inc.).

When using the measured H$_2$SO$_4$ monomer or n-mer (n = 2, 3, 4) concentrations, the concentrations of clusters containing the same H$_2$SO$_4$ molecules are summed up and written as [(H$_2$SO$_4$)$_{n,tot}$] because amines may detach from H$_2$SO$_4$-

amine clusters during the ionization process imposed by the instrument. For instance, an H$_2$SO$_4$ dimer refers to a cluster containing two H$_2$SO$_4$ molecules regardless of its base number.

The loss rates of gaseous precursors and clusters onto particles, i.e., condensation sinks (CS) and coagulation sinks, respectively, were calculated using the measured aerosol size distributions (Kulmala et al., 2001). The reported CS was calculated for H$_2$SO$_4$. The coagulation sinks of clusters and particles are usually characterized by CS or the Fuchs surface

area (McMurry et al., 2005). From the perspective of molecular kinetics, we do not distinguish coagulation and condensation in this study. Hence, CS is used to characterize the condensation and coagulation scavenging effects of aerosols on gaseous precursors, clusters, and new particles. The formation rate of 1.4 nm (in electrical mobility diameter) particles, $J_{1.4}$, is calculated using a population balance formula (Cai and Jiang, 2017). This formula improves the estimation of particle coagulation scavenging compared to previous formulae.

The uncertainty of the measured aerosol size distributions was estimated to be ±10% (Wiedensohler et al., 2012) and +100%/−50% (Kangasluoma et al., 2020) for particles larger and smaller than 10 nm, respectively. The CS in urban Beijing is mainly contributed by accumulation mode particles (Cai et al., 2017b), hence the uncertainty of CS was estimated to be ±10%. The formation rate is mainly determined by the product of new particle concentration and CS in urban Beijing (Cai and Jiang, 2017), hence the uncertainty of the measured $J_{1.4}$ was estimated to be +100%/−50%. The uncertainty of the

measured H$_2$SO$_4$ concentration is +100%/−50% according to Fig. S1 and previous studies (Kürten et al., 2012; Jokinen et al., 2012). The uncertainty of the measured amine concentrations is estimated to be similar to that of H$_2$SO$_4$ concentration.

The occurrence of NPF was determined according to the evolution of the measured aerosol size distributions. A day was classified as an NPF day if a clear new particle formation and growth pattern was observed. If no NPF event occurs on a day, it was classified as a non-event day. The rest of the days, mainly with weak NPF events that are difficult to distinguish,

were classified as undefined-days. From January 2018 to March 2019, the frequencies of NPF days and undefined days are 35% and 5%, respectively.


## 3 Model

A kinetic model was used to illustrate the nucleation process of $H_2SO_4$-amine clusters. Similar models have been reported in previous studies (Chen et al., 2012; McGrath et al., 2012; Jen et al., 2014). The cluster evaporation rate used in the model

was based on quantum chemistry calculations but modified to fit the experimental data. The standard molar Gibbs free energy of formation of $(H_2SO_4)_1(DMA)_1$, $\Delta_f G_{m,A1B1}^{\theta}(298.15\ K)$ was assumed to be -14.0 kcal·mol$^{-1}$ in this study, which is in the range of values reported in previous studies (Ortega et al., 2012; Myllys et al., 2019). The quantum chemistry results for $\Delta_f G_{m,A1B1}^{\theta}(298.15\ K)$ using the ωB97X-D/6-31++G$^{**}$, CBS-QB3, and RICC2B3 level of theory were reported to be -13.5 (Myllys et al., 2019), 14.4, and -15.4 kcal/mol (Ortega et al., 2012), respectively. The evaporation rate of $(H_2SO_4)_1(TMA)_1$ is

assumed to be 5 times that of $(H_2SO_4)_1(DMA)_1$ according to previous experimental results (Jen et al., 2014). Since the measured TMA concentration was usually lower than or comparable to the measured DMA concentration in this study, the uncertainty of the evaporation rate of $(H_2SO_4)_1(TMA)_1$ does not significantly affect the simulated particle formation rate or cluster concentrations. The value of free energy at different temperatures is calculated using Eq. S15. The evaporation rate of a cluster was derived from its corresponding standard molar Gibbs free energy of formation (Ortega et al., 2012).

140       In the proposed kinetic model, the evaporation rates of other $(H_2SO_4)_n(amine)_n$ clusters (n = 2, 3, 4) are negligible compared to their coagulation sinks and hence not included. The very unstable clusters, e.g., $(H_2SO_4)_1(amine)_3$, are omitted. Some studies indicate that other $H_2SO_4$-amine clusters, e.g., $(H_2SO_4)_3(amine)_2$, $(H_2SO_4)_3(amine)_4$ and $(H_2SO_4)_4(amine)_3$, and their corresponding reaction pathways may contribute to NPF (McGrath et al., 2012; Olenius et al., 2017), but these studies are not consistent with each other due to the uncertainties in quantum chemistry calculation. Within the ranges of these

predicted Gibbs free energies of formation, it is similarly arbitrary to assume negligible or high evaporation rates of these clusters. The kinetic model used in this study does not include the reaction pathways via these clusters (see the SI), i.e., it assumes infinitely high evaporation rates of these clusters. Ion-induced nucleation is neglected in this model because it has a minor contribution to $H_2SO_4$-amine nucleation for typical ambient conditions in the polluted environment (Yao et al., 2018).

       The cluster concentrations and particle formation rates were simulated using the kinetic model. Similarly to previous

studies (Jen et al., 2014), we assume the $A_4B_4$ formation rate as the simulated particle formation rate, $J_{1.4}$. Currently, the knowledge of the exact size of $H_2SO_4$-amine clusters is still limited. Previous studies reported that the electrical mobility of $[HSO_4(H_2SO_4)_3(DMA)_3]^-$ (Jen et al., 2015) and $[HSO_4(H_2SO_4)_6(DMA)_4]^-$ (Thomas et al., 2016) is $1.0\times10^{-4}$ and $9.4\times10^{-5}$ m$^2$·V$^{-1}$·s$^{-1}$, respectively. According to these values, the electrical mobility diameter of an $(H_2SO_4)_4(DMA)_4$ cluster was estimated to be ~1.4 nm, which locates within the measurement range of the aerosol size spectrometer. Its geometric

diameter is estimated to be ~1.1 nm according to the relationship between geometric diameter and electrical mobility diameter (Ku and de la Mora, 2009; Larriba et al., 2011).

       The uncertainty of the model mainly comes from the uncertainly in the evaporation rate of $(H_2SO_4)_1(amine)_1$. We estimated this uncertainty range using one high and one low evaporation rate from the values reported in the literature



(Ortega et al., 2012; Myllys et al., 2019). According to this estimation, the uncertainty of the model is of the same order of

magnitude as the measurement uncertainties (Fig. S2).

## 4 Results and Discussion

### 4.1 Kinetic nucleation in the presence of a high aerosol concentration

During this measurement, $H_2SO_4$-amine nucleation was found to be the dominating nucleation mechanism in the polluted atmosphere in urban Beijing. As shown in Fig. 1, the $H_2SO_4$ dimer concentration was simulated at the median values of CS

(0.017 $s^{-1}$), $C_2$-amine concentration (1.8 ppt), and temperature (281 K). Good consistency ($R^2$=0.75) was observed between the measured and simulated $H_2SO_4$ dimer concentrations. The dimer concentration is presented here because it can be reliably measured and its concentration contributes to understanding the reaction pathways (Chen et al., 2012; Jen et al., 2014; Kürten et al., 2015).

The measured $H_2SO_4$ trimer and tetramer concentrations provide further evidence for the kinetic nucleation mechanism

of $H_2SO_4$-amine clusters in urban Beijing. Figure 2 shows that the measured $H_2SO_4$ dimer, trimer, and tetramer concentrations are in accordance with their corresponding simulated concentrations when considering the uncertainties in determining the detection efficiencies of $H_2SO_4$ trimer and tetramer concentrations. The systematic difference that the measured $H_2SO_4$ trimer and tetramer concentrations are lower than the simulated concentrations is presumably caused by measurement uncertainties, e.g., cluster fragmentation and a declining detection efficiency with the increasing cluster size. In

a CLOUD chamber study on kinetic $H_2SO_4$-DMA nucleation (Kürten et al., 2014) with a high DMA concentration (5-32 ppt), such differences between simulated and measured $H_2SO_4$ n-mer concentrations were also observed. If the $H_2SO_4$ n-mer concentrations were overestimated in the kinetic model, the measured particle formation rate should also be lower than the simulated rate, which is inconsistent with the results shown in Fig. 3.

Besides the concentrations of $H_2SO_4$ clusters, there is also a consistency between the measured and simulated formation

rate of 1.4 nm (electrical mobility diameter) particles (Figs. 3 and S3). This consistency indicates that the clustering of $H_2SO_4$ and amine is the governing mechanism for nucleation and the initial growth of new particles up to 1.4 nm in urban Beijing. To compare the formation rates measured at different CS, the measured $J_{1.4}$ in Fig. 3a were scaled to the median CS (0.017 $s^{-1}$) during the observed NPF events in this study. The scaling formula is $J_{1.4,\text{scaled}} = J_{1.4} \times (\text{CS} / 0.017 \text{ s}^{-1})^2$, which is based on the fact that under the high CS, $J_{1.4}$ is theoretically inversely proportional to $CS^2$. Similarly, the measured $J_{1.4}$ in Fig.

3b were scaled by $J_{1.4,\text{scaled}} = J_{1.4} \times ((3.5 \times 10^6 \text{ cm}^{-3}) / [(H_2SO_4)_{1,\text{tot}}])^4$ to compare the formation rates measured at different $H_2SO_4$ monomer concentrations. The derivations for these scaling formulae are detailed in the SI. The dependency of measured $J_{1.4}$ on $H_2SO_4$ monomer concentrations in Fig. 3a provides supports for the scaling in Fig. 3b and vice versa.

The measured particle formation rate is then compared to previous studies. The CLOUD study reported that particle formation rate for $H_2SO_4$-DMA nucleation (red curve in Fig. 3) was obtained at a high DMA concentration (5-32 ppt), a low

cluster loss rate (Kürten et al., 2014; Kürten et al., 2018). The wall loss and dilution rates in that study sum up to be ~0.002 $s^-$


[1]. The particle formation rate under the same $H_2SO_4$ monomer concentration measured in these CLOUD experiments deviates from the measured formation rate in urban Beijing, and the reason for this deviation will be discussed in section 4.2 below. The curves from other previous studies are simulated using their reported equations (Chen et al., 2012; Jen et al., 2014; Hanson et al., 2017) and the parameters measured in this study. Some of these studies reported higher evaporation

rates of $H_2SO_4$-amine clusters according to their experimental data (Chen et al., 2012; Jen et al., 2014). However, the simulated particle formation rates using these models and evaporation rates are orders of magnitude lower than the measured particle formation rates in urban Beijing.

In addition to $H_2SO_4$ nucleation with amines, the nucleation of oxidized organics with low vitalities was also reported in the atmosphere (e.g., Bianchi et al., 2016). Various organic vapors were observed in urban Beijing (Fig. S4). However, there

was a considerable discrepancy between the absolute value and diurnal trend of particle formation rate contributed by organics and those obtained by measurements (Figs. S5 and S6), indicating that oxidized organics nucleation is not a governing mechanism to initialize NPF in urban Beijing during this campaign.

The consistency between the measured particle formation rate and the kinetic model also provides hints on the sticking probability between $H_2SO_4$-amine clusters and particles. In a previous study (Kulmala et al., 2017), it was discussed that

other condensable vapors in addition to $H_2SO_4$ and amine may contribute to the initial growth of new particles and that the coagulation scavenging effect in the polluted environment may be overestimated because of the overestimated sticking probability between particles or clusters. However, in this study, we found that up to ~1.4 nm particles (in electrical mobility diameter), the particle formation rate estimated using $H_2SO_4$-amine clustering and a sticking probability of 1.0 is consistent with the measured formation rate. The measured $H_2SO_4$ trimer and tetramer concentrations are even lower than the simulated

concentrations (Fig. 2) due to potential measurement uncertainties, whereas a significant contribution of other condensable vapors or an overestimated coagulation sink will theoretically result in higher measured concentrations compared to the simulated concentrations. However, the further growth of particles beyond ~1.4 nm in polluted environments still needs further explorations.

## 4.2 The influence of coagulation scavenging

The scavenging of $H_2SO_4$-amine clusters due to coagulation with larger particles is a major limiting factor for NPF in urban Beijing. After normalizing the influence of $H_2SO_4$ monomer concentration, the particle formation rate decreases with an increasing CS (Fig. 3b). Negative dependencies of $H_2SO_4$ cluster concentration and sub-3 nm particle concentration on CS were also reported in our previous studies in urban Beijing (Cai et al., 2017b; Deng et al., 2020a). However, there was usually a good positive correlation between CS and amine concentrations in urban Beijing, presumably due to the correlation between their sources. As a result, the apparent dependency between the measured $J_{1.4}$ and CS in Fig. 3b was also influenced

by amine concentrations.

Although coagulation scavenging does not affect the detailed equilibrium of reactions, it can have significant impacts on the steady-state cluster concentrations. For instance, the particle formation rate of the CLOUD chamber experiments





deviates from the measured and simulated formation rates in urban Beijing. This indicates that although the $H_2SO_4$
concentration in these chamber studies was in the typical ambient range, these results from chamber experiments are not
directly applicable to represent the real atmospheric conditions in urban Beijing due to the difference in coagulation
scavenging rates characterized by CS. The median CS in urban Beijing during the NPF events in this field measurement was
~0.017 $s^{-1}$, which is nearly an order of magnitude higher than the total loss rates in the chamber studies (Kürten et al., 2014;
Kürten et al., 2018; Hanson et al., 2017). Hence, the curve for the CLOUD chamber experiments in Fig. 3 deviates from the
measured data in urban Beijing.

       The power of particle formation rate to $H_2SO_4$ monomer concentration, $[(H_2SO_4)_{1,tot}]^p$ is consistent with the argument
that coagulation scavenging is a limiting factor for NPF in the polluted environment. It can be proven that under the CS-
controlled regime, the power of the formation rate of $(H_2SO_4)_4(amine)_4$ to $H_2SO_4$ concentration, $p$, is ~4.0 rather than 2.0
(Fig. S7), which is consistent with the measured $p$ in urban Beijing (Fig. 3a). In the perspective of conventional kinetic
nucleation theory, the critical step of nucleation is the formation of $H_2SO_4$ dimer clusters. Accordingly, the $p$-value is
expected to be 2.0 (Kuang et al., 2008). This power dependency was also used to prove that there was no significant
evaporation of $(H_2SO_4)_n(amine)_n$ for n > 2 in a previous CLOUD chamber study (Kürten et al., 2014). However, this theorem
is valid only when the external cluster losses are negligible (Ehrhart and Curtius, 2013; Kupiainen-Määttä et al., 2014; Elm
et al., 2020), whereas in the presence of a high aerosol concentration, the loss rates of $H_2SO_4$-amine clusters are usually an
order of magnitude higher than their growth rates into large clusters. As a result, the cluster concentrations maintain a
pseudo-steady state and their growth fluxes into the next larger clusters are proportional to, rather than independent of,
$H_2SO_4$ concentration (as detailed in the SI).

### 4.3 The influence of amine concentrations

       In addition to coagulation scavenging, the low effective amine concentration is another limiting factor for NPF in urban
Beijing. During the measurement period in urban Beijing, the median $C_2$-amine concentrations for the daytime NPF period
and all the observation periods were 1.8 and 2.7 ppt, respectively. Meanwhile, measured $H_2SO_4$ dimer concentration and
particle formation rate in urban Beijing were lower than the amine-saturation limit (Figs. 1 and 3). Amine-saturation means
that further increasing the amine concentrations does not significantly enhance the nucleation rate. In contrast, under
unsaturated amine concentrations, with respect to the formation of $H_2SO_4$-amine clusters, they are not stable against
evaporation. The measured $H_2SO_4$ dimer concentration and particle formation rate indicate a moderate evaporation rate of
$H_2SO_4$-amine clusters because they were close to but lower than their corresponding amine-saturation limits. This
unsaturated particle formation rate with a low effective amine concentration is consistent with the saturation concentration of
amines reported in previous chamber experiments (Almeida et al., 2013; Jen et al., 2014).

       In addition, the dependency of the measured particle formation rate on amine concentrations in Fig. 3a provides support
for the view that unsaturated $H_2SO_4$-amine nucleation occurs in urban Beijing. Note that the apparent correlation between the
effective amine concentration and CS was minimized in Fig. 3a by scaling the measured $J_{1.4}$ with respect to CS. In contrast,





the apparent negative correlation between the measured $J_{1.4}$ and the effective amine concentration in Fig. 3b is governed by the positive correlation between the effective amine concentration and CS in urban Beijing. For the same reason, a negative correlation between NPF and amine concentrations was also reported in central Germany (Kürten et al., 2016).

In the above analysis, DMA is thought to be a major base that stabilizes $H_2SO_4$-amine clusters and TMA may also contribute. Other bases, e.g., monomethylamine and $NH_3$, were measured but are not included in our analysis due to their relatively weak bond to $H_2SO_4$ molecules. Although $NH_3$ concentrations are high (with a median value of 789 ppt during the observed NPF in this study), theoretically $NH_3$ cannot be the major base to stabilize $H_2SO_4$ in urban Beijing due to the high evaporation rate of the $NH_4HSO_4$ molecule (Ortega et al., 2012; Jen et al., 2014; Olenius et al., 2017). However, these
relatively weak bases may contribute to the particle growth and their synergistic effects and base substitutions have been reported in previous studies (Kupiainen et al., 2012; Glasoe et al., 2015; Myllys et al., 2019). $C_1$- and $C_4$-amines stabilized neutral $H_2SO_4$ trimers are detected during NPF events in this study, as shown in Fig. S4a. Despite these potential contributions, the formation of large clusters until $(H_2SO_4)_4(amine)_4$ and ~1.4 nm particles (in electrical mobility diameter) in urban Beijing can be quantitatively explained by the kinetic model (Figs. 1, 2, 3, and S5). Some other compounds in addition
to amines, such as diamines (Jen et al., 2016) and guanidine (Myllys et al., 2019), are also reported to be possible bases to stabilize $H_2SO_4$ clusters; however, they were not observed during this measurement.

### 4.4 The synergistic influences of $H_2SO_4$, coagulation scavenging, and amine concentrations

    Summarizing the synergistic effect of $H_2SO_4$ monomer concentration, CS, and the effective amine concentration, the governing factors for $H_2SO_4$-amine nucleation at different regimes are illustrated in Fig. 4. The horizontal coordinate is equal
to $0.5\beta_{11}\cdot[(H_2SO_4)_{1,tot}]/CS$, where $\beta_{11}$ is the collision coefficient of two $(H_2SO_4)_1(amine)_1$ clusters. It characterizes the ratio of the condensational growth rate of a molecule or cluster to its loss rate. The vertical coordinate is equal to $\beta_{AB}[amine]/(CS+\gamma)$, where $\beta_{AB}$ is the collision coefficient of an $H_2SO_4$ molecule and an amine molecule and $\gamma$ is the evaporation rate of $(H_2SO_4)_1(amine)_1$. As indicated by this formula, this vertical coordinate is linearly proportional to the effective amine concentration. $J_{1.4}$ was estimated using the proposed model and it is normalized by dividing it by its collision limit ($J_c$) at the
same $H_2SO_4$ monomer concentration. The collision limit refers to the collision rate of two $H_2SO_4$ monomers, which is theoretically the maximum steady-state particle formation rate. The effective amine concentration is assumed to be independent of $[(H_2SO_4)_{1,tot}]$, i.e., it is assumed that the formation of $(H_2SO_4)_1(amine)_1$ does not cause a change in amine concentrations. It should be clarified that due to evaporation and the minor reaction pathways, the normalized particle formation rate is governed but not only determined by the normalized sulfuric and amine concentrations. Hence, the
background color map shown in this figure is illustrative rather than quantitative.

    According to the region of measured data in Fig. 4, particle formation rate is sensitive to both CS and amine concentrations in the real atmosphere of urban Beijing and Nanjing, whereas it is insensitive to amine concentrations in most of the experimental conditions in the CLOUD studies (Almeida et al., 2013; Kürten et al., 2014). A high $C_2$-amine concentration was reported for Shanghai (40 ± 14 ppt, Yao et al., 2016) and therefore the effective amine concentration in





Fig. 4 locates in a similar range to that in the CLOUD experiments. In the upper right corner of Fig. 4, both the sulfuric and amine concentrations are sufficiently high so that the steady-state cluster and particle formation rates are governed by the collision rate of two $H_2SO_4$ monomers. At the upper left corner, nucleation is controlled by CS because the concentrations of clusters and particles are governed by both their formation and loss rates (see Eqs. S22 – S28 in the SI). In this CS-controlled regime, increasing the effective amine concentration in a narrow range does not significantly increase the formation rate

because the effective amine concentration is sufficient with respect to the evaporation of $H_2SO_4$-amine clusters and the formation rate is close to its amine-saturation limit. At the lower right corner, the formation and growth of $H_2SO_4$-amine clusters are not limited by their coagulation losses. However, due to the low effective amine concentration, considerable evaporation of $H_2SO_4$-amine clusters limits the formation rate of new particles.

## 5 Conclusions

The predominating mechanism initiating NPF in urban Beijing is illustrated. Compared to previous studies that investigated $H_2SO_4$-amine nucleation under high amine concentrations in laboratories (Almeida et al., 2013; Jen et al., 2014; Kürten et al., 2014) and reported it to be a governing mechanism in the atmosphere (Yao et al., 2018), we further show the governing roles of $H_2SO_4$, amines, and the coagulation scavenging effect at the molecular level based on the long-term ambient atmospheric measurements in urban Beijing. Comparing the measured particle formation rate and cluster concentrations with those

simulated using a kinetic model, we demonstrated these new findings. The formation and growth of $H_2SO_4$-amine clusters under the strong coagulation scavenging effect seems to be a major pathway for cluster growth up to ~1.4 nm particles (in electrical mobility diameter). Both theoretical analysis and measured data support that differently from previous chamber studies (Kürten et al., 2014; Almeida et al., 2013) and atmospheric measurements (Yao et al., 2018), the typical amine concentrations measured in this study are sometimes insufficient to bound with nearly all the $H_2SO_4$ monomers into $H_2SO_4$-

amine clusters. The sensitivity of NPF to amine concentrations also indicates that the contributions of NPF to the aerosol number and surface concentrations will decrease if atmospheric amine concentrations are reduced. Due to the correlated variables and measurement uncertainties in atmospheric measurements, the quantitative influences of various amines, water vapor, and coagulation sink on NPF need further verifications from experiments in chambers or other controlled systems. For future chamber studies, we recommend that the gaseous precursors and condensation sink should be at their typical

ambient values, as their values not only affect particle formation rate, but also the detailed nucleation kinetics.

*Author contributions*. R.C. and J.J. designed the research and wrote the paper with inputs from other co-authors; C.Y., F.B., Y. Liu, L.W., J.Z., M.K., and J.J. contributed to designing measurement station; R.C., C.Y., D.Y., R.Y., Y. Lu, C.D., Y. F., J.R., X.L., Q.Z., and J. Kangasluoma contributed to data collection; R.C., C.Y., D.Y., R.Y., and J.J. analyzed data with the

help from Y.Lu, C.D., Y.F., J.R., X.L., J. Kontkanen, Q.Z., J. Kangasluoma, Y.M., J.H., D.R.W., F.B., P.P., V.-M.M., Y. Liu, L.W., J.Z., and M.K.



*Code availability*. The Julia code for the kinetic model is available upon request.

*Data availability: The data that support the findings of this study are available from the corresponding author upon request.*

*Competing interests*. The authors declare that they have no conflict of interest.

*Acknowledgment*. The work is supported by the National Key R&D Program of China (2017YFC0209503 & 2017YFC0209505), the National Science Foundation of China (21876094, 41730106, and 91644213), Academy of Finland (project no. 332547), Academy of Finland via Center of Excellence in Atmospheric Sciences (project no. 272041, 316114, and 315203), Samsung PM$_{2.5}$ SRP, the Doctoral Programme in Atmospheric Sciences at the University of Helsinki, and European Research Council via ATM-GTP 266 (742206).






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





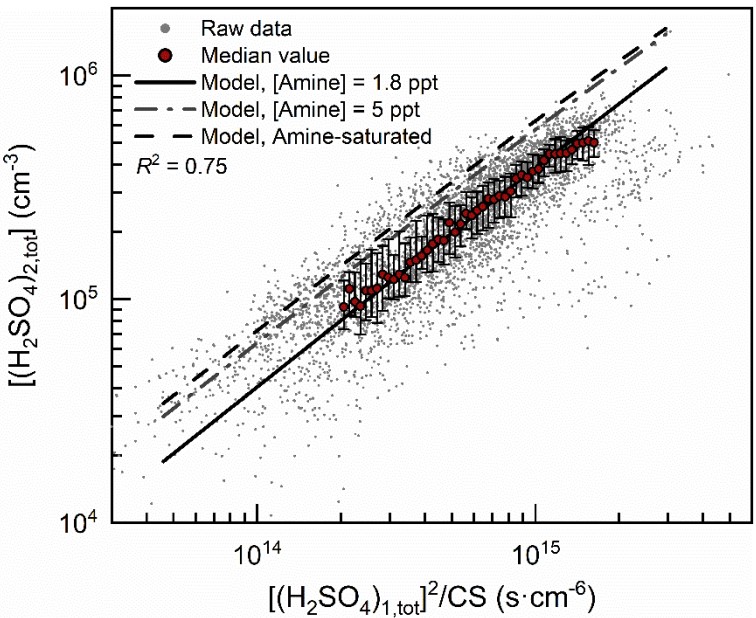

Figure 1: **H₂SO₄ dimer concentration for urban Beijing.** $[(H_2SO_4)_{1,tot}]$ and $[(H_2SO_4)_{2,tot}]$ are the measured $H_2SO_4$ monomer and dimer concentrations, respectively. The measured raw data with a temporal resolution of 5 min are shown with small dots. The median values of these dots grouped by the horizontal coordinate are shown with big red markers, and the error bars indicate the lower and upper quartiles. The condensation sink (CS) and temperature for the simulated formation rate are their median values during the observed NPF events, i.e., 0.017 s⁻¹ and 281 K, respectively. The amine-saturated limit (dashed black line) is simulated at an ultra-high DMA concentration ($10^6$ ppt) so that the evaporation rate of $H_2SO_4$-DMA clusters are negligible compared to their formation rates. The $R^2$ value was calculated using logarithmic values.





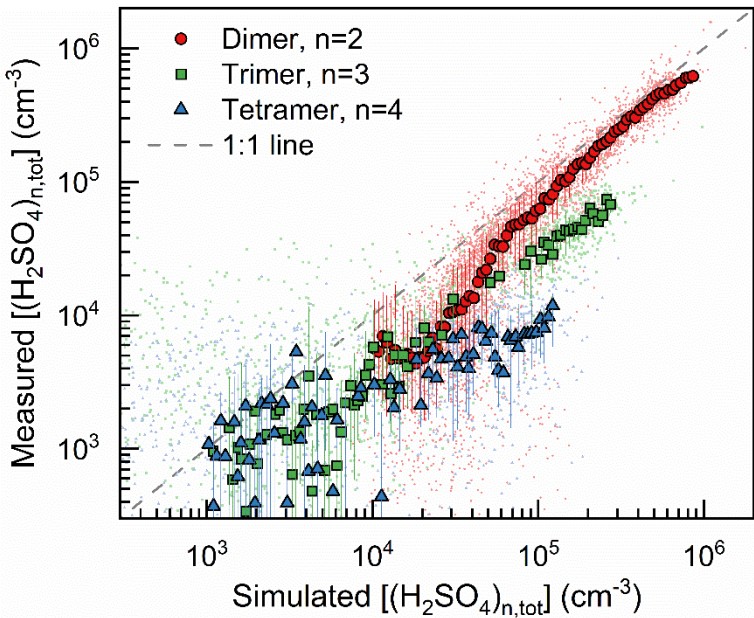

Figure 2: **The measured and simulated H₂SO₄ dimer, trimer, and tetramer concentrations.** The concentrations of clusters containing the same $H_2SO_4$ molecule number are summed together. The small dots are the raw data with a temporal resolution of 10 min and their median values are shown with big markers. The error bars indicate the lower and upper

quartiles. The uncertainty of instrument detection efficiency contributes to the difference between the measured and simulated values, as explained in the main text.





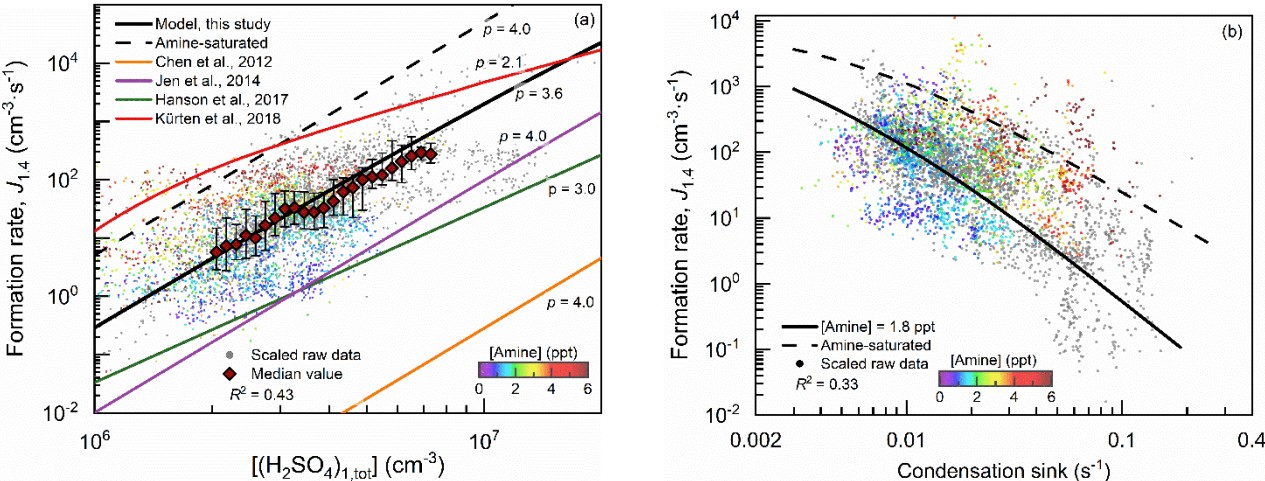

Figure 3: **The measured and simulated particle formation rates ($J_{1.4}$) as a function of (a) H₂SO₄ monomer concentration ($[(H_2SO_4)_{1,tot}]$) and (b) condensation sink (CS).** The measured raw data with a temporal resolution of 5 min are shown with small dots and colored by the measured effective amine concentration. When amine concentrations are not available, the corresponding dots are shown in grey. The median values of these dots grouped by the horizontal coordinate in (a) are shown with big red markers, and the error bars indicate the lower and upper quartiles. The measured formation rates shown in (a) and (b) are scaled to the median CS and $[(H_2SO_4)_{1,tot}]$ for the measured NPF events, respectively. The scaling formulae are given in the main text and illustrated in the SI. Note that the simulated formation rates are not scaled. $p$ is the simulated power of $J_{1.4}$ with respect to $[(H_2SO_4)_{1,tot}]$. The $R^2$ values were calculated using logarithmic values.



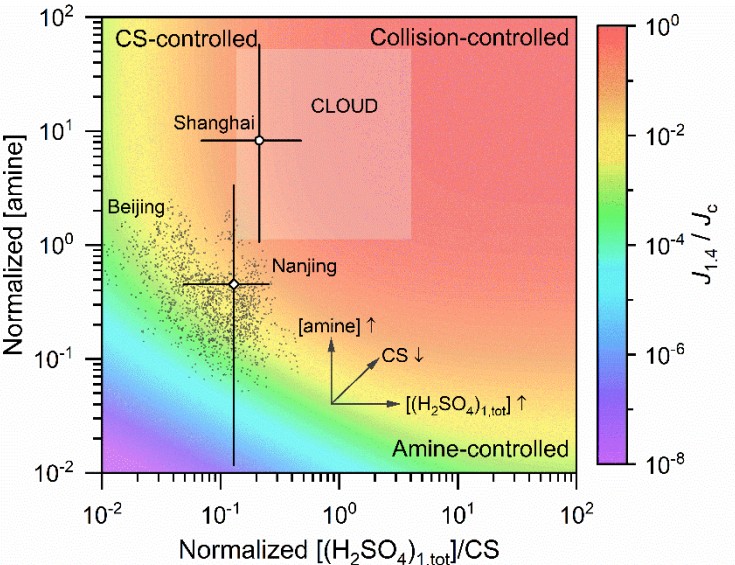

**Fig. 4. An Illustrative figure for the governing factors of H₂SO₄-amine nucleation in various conditions.** The horizontal

and vertical coordinates are the normalized $H_2SO_4$ and effective amine concentrations, respectively, and the normalizing

formulae are given in the main text. The color indicates the normalized steady-state formation rate of 1.4 nm particles, $J_{1.4}$. $J_c$

is the formation rate at the collision limit. The dark grey markers are the measured data in Beijing with a temporal resolution

of 5 min. The semi-transparent square above the colored contour is the estimated range for experimental conditions of the

CLOUD studies (Almeida et al., 2013; Kürten et al., 2014): $[(H_2SO_4)_{1,tot}]$ between $5 \times 10^5$ and $1.5 \times 10^7$ cm⁻³, [amine] between

3 and 140 ppt, $T = 278$ K, and the sum of wall loss and dilution rates (instead of CS) was estimated to be 0.002 s⁻¹. The open

markers and error bars indicate the median values and approximate ranges, respectively, of the normalized $H_2SO_4$ and

effective amine concentrations in Shanghai and Nanjing. The Shanghai data was reported by Yao et al. (2016) and Yao et al.

(2018). The Nanjing data was reported by Zheng et al. (2015) and Deng et al. (2020a). Note that for Shanghai and Nanjing,

the amine concentrations were measured in different campaigns from those for $H_2SO_4$ and CS and the amine concentrations

are the average of all days rather than new particle formation period only. As a result, there are potential uncertainties in the

results for Shanghai and Nanjing.