# Peer review of "Sulfuric acid-amine nucleation in urban Beijing"

_Atmospheric Chemistry and Physics, 2020_

## Referee Comment (RC1) · Anonymous Referee #1 · 22 Dec 2020

Cai et al. present observations of sulfuric acid and amine nucleation in Beijing, a polluted city, and determine how coagulation sink plays a role in 1.4 nm nucleation rate vs sulfuric acid concentrations. They used a suite of instruments to measure sulfuric acid, its clusters, and particle size distributions. They model nucleation based on previous acid-base chemical reaction models where the limiting step is the formation of the aminated monomer. The manuscript provides good level of detail to justify their assumptions and analysis and is easy to read. This paper should be published in this journal once they address the minor comments below

Abstract: The whole manuscript is written quite clearly except for the last two sentences of the abstract. Rewrite them so they are less colloquial and more to the point.

Page 6 line 174: Detection efficiency of the larger sulfuric acid clusters definitely lower. The authors mention that detection efficiency decreases with higher masses. Do they

mean transmission efficiency through the mass filter? If so, did the authors correct for this as shown in (Heinritzi et al., 2020; Zhao et al., 2010)? Also, the authors mention that the CIMS were calibrated for sulfuric acid concentration but how did they determine concentrations of the larger clusters? These cannot be calibrated so they must have made assumptions to calculate their concentrations from mass spec signals.

Page 7 after sentence 195. Add a sentence providing a probable reason for why simulate particle formation rates from Chen et al. and Jen et al. are lower than measured in this study.

Page 7, line 200: Despite the fact that the diurnal trends of particle formation rates do not match the concentration profiles of all observed oxidized organics, is it still possible that these organics play a role in the observed NPF despite high CS? Feels like organics encompass thousands of types of molecules and this paragraph is too dismissive of a potential role of organics in NPF.

Page 9 line 275. "It characterizes the ratio of 275 the condensational growth rate of a molecule or cluster to its loss rate." It is too vague. Can the authors change it to horizontal coordinate?

Page 10 and Line 305 in the conclusion. "Comparing the measured particle formation rate and cluster concentrations with those simulated using a kinetic model, we demonstrated these new findings." Could the authors be more specific about their new findings?

Works cited in this review: Heinritzi, M., Dada, L., Simon, M., Stolzenburg, D., Wagner, A. C., Fischer, L., Ahonen, L. R., Amanatidis, S., Baalbaki, R., Baccarini, A., Bauer, P. S., Baumgartner, B., Bianchi, F., Brilke, S., Chen, D., Chiu, R., Dias, A., Dommen, J., Duplissy, J., Finkenzeller, H., Frege, C., Fuchs, C., Garmash, O., Gordon, H., Granzin, M., El Haddad, I., He, X., Helm, J., Hofbauer, V., Hoyle, C. R., Kangasluoma, J., Keber, T., Kim, C., Kürten, A., Lamkaddam, H., Laurila, T. M., Lampilahti, J., Lee, C. P., Lehtipalo, K., Leiminger, M., Mai, H., Makhmutov, V., Manninen, H. E., Marten, R., Mathot,

S., Mauldin, R. L., Mentler, B., Molteni, U., Müller, T., Nie, W., Nieminen, T., Onnela, A., Partoll, E., Passananti, M., Petäjä, T., Pfeifer, J., Pospisilova, V., Quéléver, L. L. J., Rissanen, M. P., Rose, C., Schobesberger, S., Scholz, W., Scholze, K., Sipilä, M., Steiner, G., Stozhkov, Y., Tauber, C., Tham, Y. J., Vazquez-Pufleau, M., Virtanen, A., Vogel, A. L., Volkamer, R., Wagner, R., Wang, M., Weitz, L., Wimmer, D., Xiao, M., Yan, C., Ye, P., Zha, Q., Zhou, X., Amorim, A., Baltensperger, U., Hansel, A., Kulmala, M., Tomé, A., Winkler, P. M., Worsnop, D. R., Donahue, N. M., Kirkby, J. and Curtius, J.: Molecular understanding of the suppression of new-particle formation by isoprene, Atmospheric Chem. Phys., 20(20), 11809–11821, doi:https://doi.org/10.5194/acp-20-11809-2020, 2020. Zhao, J., Eisele, F. L., Titcombe, M., Kuang, C. and McMurry, P. H.: Chemical ionization mass spectrometric measurements of atmospheric neutral clusters using the cluster-CIMS, J Geophys Res, 115, D08205, doi:10.1029/2009jd012606, 2010.

---

## Referee Comment (RC2) · Anonymous Referee #2 · 2 Jan 2021

The H2SO4-amine nucleation is a potentially important NPF pathway in the polluted boundary layer. While the importance of this mechanism has been shown in chamber studies and in certain megacities, whether this mechanism plays an important role in cities with a relatively low amine concentration and high existing aerosol concentration like Beijing remains unclear. This study combines long-term measurements at an urban site in Beijing and kinetic modeling to show that the H2SO4-amine nucleation is a dominant mechanism to initialize NPF in Beijing. The governing factors for H2SO4-amine nucleation are also elucidated. This work is meaningful for improving our understanding of NPF mechanism in polluted environments. The paper is generally well written. I think it can be accepted for publication after revisions to address the following (mostly minor) comments and suggestions.

(1) You tried to exclude organic nucleation as a main NPF pathway in Beijing (Line 198-202, Figs. S5 and S6). However, you only considered pure organic nucleation.

[Figure]

A potentially important pathway in polluted environments is the nucleation of organics with $H_2SO_4$. I think some calculations are needed to explore whether this mechanism could play a role that is comparable to the $H_2SO_4$-amine mechanism.

(2) In Fig. S5, only ELVOCs are used in the calculation of pure organic nucleation rate. I think Kirkby et al. (2016)'s equation was based on HOMs. What adjustment was made to the original equation?

(3) The opening sentence of Results and Discussion gives a major conclusion of this study. However, the relationships between this conclusion and the supporting evidence detailed below are not very clear. For example, how do the results presented in Figs. 1 and 2 support this conclusion? Although I know the underlying logic, it is unfortunately not explained in the paper. The connections between the evidence and the conclusion should be described directly and clearly.

(4) The field measurements in Beijing were conducted from January 2018 to March 2019. However, it is not clear which parts of these data are used in the results shown in Figs. 1-4. Did you use measurements on all days or NPF days only? Is every NPF day between January 2018 and March 2019 included in these figures?

(5) I think the kinetic simulations are only done at selected conditions based on the description in Line 165. However, the measurements cover a wide range of conditions which, as you show in the paper, have a large impact. How do you make sure that you are doing an apple-to-apple comparison between modeling and measurements in Figs. 1 and 2?

(6) How are the results in Fig. 4 derived exactly? From the kinetic model or a combination of model and measurements?

(7) Line 167: The sentence is vague. How does the dimer concentration contribute to understanding the reaction pathways?

(8) Line 195-197: The simulated particle formation rates using these previous models

and evaporation rates are orders of magnitude lower than the measured particle formation rates in urban Beijing. Why does this happen? Does this affect the robustness of your conclusion?

(9) Line 251-253: You may want to directly give the saturation concentration of amines here.

(10) Line 288-290: What are the main reasons for the much higher amine concentration in Shanghai than that in Beijing?
* * *

---

## Author Comment (AC1) · 11 Jan 2021

(Sulfuric acid-amine nucleation in urban Beijing)

We thank two anonymous reviewers for their efforts and constructive comments that help to improve this manuscript. The reviewer's comments are addressed in the following paragraphs and the manuscript and the supplementary information were revised minorly. The comments are shown as sans-serif dark red texts and our responses are shown as serif black texts. Changes are highlighted in the revised manuscript and shown as "quoted underlined texts" in the responses. References are given at the end of the responses.

**Reviewer #1**

Cai et al. present observations of sulfuric acid and amine nucleation in Beijing, a polluted city, and determine how coagulation sink plays a role in 1.4 nm nucleation rate vs sulfuric acid concentrations. They used a suite of instruments to measure sulfuric acid, its clusters, and particle size distributions. They model nucleation based on previous acid-base chemical reaction models where the limiting step is the formation of the aminated monomer. The manuscript provides good level of detail to justify their assumptions and analysis and is easy to read. This paper should be published in this journal once they address the minor comments below.

Abstract: The whole manuscript is written quite clearly except for the last two sentences of the abstract. Rewrite them so they are less colloquial and more to the point.

**Response**: Thanks. These sentences were revised as "The formation of $H_2SO_4$-amine clusters in Beijing is sometimes limited by low amine concentrations. Summarizing the synergistic effects of $H_2SO_4$ concentration, amine concentrations, and aerosol concentration, we elucidate the governing factors for $H_2SO_4$-amine nucleation for various conditions."

Page 6 line 174: Detection efficiency of the larger sulfuric acid clusters definitely lower. The authors mention that detection efficiency decreases with higher masses. Do they mean transmission efficiency through the mass filter? If so, did the authors correct for this as shown in (Heinritzi et al., 2020; Zhao et al., 2010)? Also, the authors mention that the CIMS were calibrated for sulfuric acid concentration but how did they determine concentrations of the larger clusters? These cannot be calibrated so they must have made assumptions to calculate their concentrations from mass spec signals.

**Response**: The mass-dependent transmission efficiency was corrected. The calibration factors for larger clusters were assumed to be equal to the factor for $H_2SO_4$ molecules because of lacking calibration standards. We added a few sentences in Section 2 for clarification:

"In additions to $H_2SO_4$ molecules, $H_2SO_4$ clusters and organics with low volatilities were measured using ToF-CIMS. Their calibration factors were assumed to be equal to that for $H_2SO_4$ molecules. The mass dependency of transmission efficiency was calibrated and corrected using the method reported in Heinritzi et al. (2016)."

In addition, this sentence on Page 6 was revised as "...... is presumably caused by measurement uncertainties, e.g., cluster fragmentation (Zapadinsky et al., 2019)".

Page 7 after sentence 195. Add a sentence providing a probable reason for why simulate particle formation rates from Chen et al. and Jen et al. are lower than measured in this study.

**Response**: This sentence was revised as "However, due to these high evaporation rates, the simulated particle formation rates using these models are orders of magnitude lower than the measured particle formation rates in urban Beijing." This revised sentence emphasizes that the high evaporation rate is the main cause of the low formation rate.

Page 7, line 200: Despite the fact that the diurnal trends of particle formation rates do not match the concentration profiles of all observed oxidized organics, is it still possible that these organics play a role in the observed NPF despite high CS? Feels like organics encompass thousands of types of molecules and this paragraph is too dismissive of a potential role of organics in NPF.

**Response**: The absolution concentration of ELVOCs (or HOMs) measured in this study is insufficient to explain the measured high formation rate of 1.4 nm particles, as shown in Fig. S5 and discussed in its caption. Meanwhile, the measured absolution values and diurnal trends of particle formation rates were consistent with those simulated using the $H_2SO_4$-amine nucleation mechanism. Hence, we concluded that "oxidized organics nucleation is not a governing mechanism to initialize NPF in urban Beijing during this campaign".

However, organics may contribute to particle growth and hence play an important role in NPF. With an increasing aerosol size and due to a potential effect of the Rauolt's law, organics may condense onto $H_2SO_4$-amine clusters and enhance their subsequent growth. A previous study (Deng et al., 2020b) in urban Beijing reported a significant positive size dependence of new particle growth rate. The contributions from the $H_2SO_4$, amine, $NH_3$, and $H_2SO_4$-amine clusters were found to be insufficient to explain the measured growth rate for particles larger than 3 nm. This discrepancy indicate the important roles of other species, including organics, in the subsequent growth of new particles. To avoid confusion, we added "Note that organics with low volatilities may contribute to the growth of larger (e.g., > 2 nm) particles (Deng et al., 2020b)" to the end of this paragraph.

Page 9 line 275. "It characterizes the ratio of the condensational growth rate of a molecule or cluster to its loss rate." It is too vague. Can the authors change it to horizontal coordinate?

**Response**: We added formulae to clarify the growth rate and loss rate in this sentence and help to read to understanding the physical meaning of the horizontal coordinate in Fig. 4. This sentence was revised as "It characterizes the ratio of the condensational growth rate $(\underline{\beta_{11} \cdot [(H_2SO_4)_{1,tot}]})$ of a molecule or cluster to its loss rate (CS)."

Page 10 and Line 305 in the conclusion. "Comparing the measured particle formation rate and cluster concentrations with those simulated using a kinetic model, we demonstrated these new findings." Could the authors be more specific about their new findings?

**Response**: This sentence was revised as "Comparing the measured particle formation rate and cluster concentrations with those simulated using a kinetic model, we demonstrated and quantified the influences of $H_2SO_4$ concentration, amine concentration and coagulation scavenging on new formation rate in urban Beijing."

**Reviewer #2**

The $H_2SO_4$-amine nucleation is a potentially important NPF pathway in the polluted boundary layer. While the importance of this mechanism has been shown in chamber studies and in certain megacities, whether this mechanism plays an important role in cities with a relatively low amine concentration and high existing aerosol concentration like Beijing remains unclear. This study combines long-term measurements at an urban site in Beijing and kinetic modeling to show that the $H_2SO_4$-amine nucleation is a dominant mechanism to initialize NPF in Beijing. The governing factors for $H_2SO_4$-amine nucleation are also elucidated. This work is meaningful for improving our understanding of NPF mechanism in polluted environments. The paper is generally well written. I think it can be accepted for publication after revisions to address the following (mostly minor) comments and suggestions.

(1) You tried to exclude organic nucleation as a main NPF pathway in Beijing (Line 198-202, Figs. S5 and S6). However, you only considered pure organic nucleation. A potentially important pathway in polluted environments is the nucleation of organics with $H_2SO_4$. I think some calculations are needed to explore whether this mechanism could play a role that is comparable to the $H_2SO_4$-amine mechanism.

**Response**: We agree with the reviewer that $H_2SO_4$-organics nucleation may be an important nucleation mechanism in some atmospheric environments (e.g., Lehtipalo et al., 2018). However, similar to the comparison between simulated and measured new particle formation rates for pure organics nucleation mechanism in Fig. S5, the new particle formation rate driven by the $H_2SO_4$-organics nucleation mechanism (e.g., reported in Riccobono et al., 2014) is more than one order of magnitude lower than the measured formation rates in this study. For instance, at $[(H_2SO_4)_{1,tot}] = 5 \times 10^6$ cm$^{-3}$, the new formation rates measured in urban Beijing and reported in Riccobono et al. (2014) were ~100 cm$^{-3} \cdot$s$^{-1}$ and <10 cm$^{-3} \cdot$s$^{-1}$, respectively. The discrepancy between these two formation rates is supposed to be even larger because of the high coagulation sink and low concentrations of ELVOCs (or HOMs) in urban Beijing.

(2) In Fig. S5, only ELVOCs are used in the calculation of pure organic nucleation rate. I think Kirkby et al. (2016)'s equation was based on HOMs. What adjustment was made to the original equation?

**Response**: The ELVOCs concentration is taken as HOMs concentration for this estimation. We clarified this in the revised caption of Fig. S5.

Kirkby et al. (2016) show that at a low HOMs concentration (which is the case for urban Beijing), the new particle formation rate driven by pure organic nucleation is mainly contributed by the ion-induced pathway. Due to the limited ion production rate and high condensation sink, the formation rate contributed by pure organic nucleation rate is not comparable to the measured high formation rate in urban Beijing.

(3) The opening sentence of Results and Discussion gives a major conclusion of this study. However, the relationships between this conclusion and the supporting evidence detailed below are not very clear. For example, how do the results presented in Figs. 1 and 2 support this conclusion? Although I know the underlying logic, it is unfortunately not explained in the paper. The connections between the evidence and the conclusion should be described directly and clearly.

**Response**: We added a paragraph right after this opening sentence to elucidate the logic of this study:

"This finding is supported by comparing the measured and simulated $H_2SO_4$ cluster concentrations and new particle formation rates. The consistency between the measurement results and the $H_2SO_4$-amine nucleation mechanism is shown and discussed below. Meanwhile, other nucleation mechanisms, e.g., $H_2SO_4$-$NH_3$ nucleation and organics nucleation, were found to be not sufficient to explain the observed high new particle formation rate under the high coagulation sink (see Section 4.3 and the supporting information)."

(4) The field measurements in Beijing were conducted from January 2018 to March 2019. However, it is not clear which parts of these data are used in the results shown in Figs. 1-4. Did you use measurements on all days or NPF days only? Is every NPF day between January 2018 and March 2019 included in these figures?

**Response**: Dates with available data and the parts of data used in Figs. 1-4 were clarified in the revised manuscript.

The $H_2SO_4$ concentration was not available in the summer of 2018. Since the $H_2SO_4$ concentration was used in Figs. 1-4 and the aerosol size distribution data was mostly available during this campaign, we report the measurement period of this study based on the availability of the $H_2SO_4$ concentration. The sentence on the measurement period in Section 2 was revised as "The dataset for this study is from January to April 2018 and October 2018 to March 2019."

Figures 1, 3, and 4 use the data on NPF days only. Figure 2 uses data on both NPF days and non-NPF days because $H_2SO_4$ clusters can also be measured on non-NPF days. This information was added to the captions of figures in the revised manuscript.

(5) I think the kinetic simulations are only done at selected conditions based on the description in Line 165. However, the measurements cover a wide range of conditions which, as you show in the paper, have a large impact. How do you make sure that you are doing an apple-to-apple comparison between modeling and measurements in Figs. 1 and 2?

**Response**: We thank the reviewer for pointing this out. Case-to-case comparison between the measured and simulated data is included in this manuscript. The simulation conditions were determined according to the axes of each figure. When comparing two variables, the simulation condition was selected as the median value of the measured conditions, and the variations of other variables were minimized by scaling in some figures. However, when comparing the measured and simulated values in Figs. 2, S3, and S5, the simulation condition was case-specific. We added a sentence to the caption of Fig. 2: "The simulation condition was case-specific for each raw data dot."

(6) How are the results in Fig. 4 derived exactly? From the kinetic model or a combination of model and measurements?

**Response**: The results shown with markers in Fig. 4 were obtained via the combination of measurements, theory, and the kinetic model. The $H_2SO_4$ concentration, amine concentration, and condensation sink were measured values. The evaporation rate was calculated with simulated Gibbs free energy, simulated enthalpy, and measured temperature. The formation rate at the collision limit is by definition equal to the collision rate between $H_2SO_4$ monomers. The formation rate shown by the background color map was simulated using the kinetic model.

The caption of Fig. 4 was revised to illustrate the calculation of the results: "The color indicates the normalized steady-state formation rate of simulated 1.4 nm particles, $J_{1.4}$. $J_c$ is the theoretical formation rate at the collision limit. The dark grey markers are the measured data on NPF days in Beijing with a temporal resolution of 5 min."

(7) Line 167: The sentence is vague. How does the dimer concentration contribute to understanding the reaction pathways?

**Response**: The $H_2SO_4$ dimer concentration is closely related to new particle formation for three reasons: 1) $H_2SO_4$ dimer is an intermediate of the nucleation process; 2) Both experiments (Kürten et al., 2014) and quantum chemistry simulations (Ortega et al., 2012; Myllys et al., 2019) indicates the stability of $H_2SO_4$ dimer clusters; 3) there is a mathematical coincidence between the dimer concentration and particle formation rate under a high coagulation sink.

Only 1) is used in this manuscript and it is easy to be understood by the readers. It is difficult to explain 2), 3), and how to relate $H_2SO_4$ dimer concentration to the reaction pathway within a few sentences; whereas explaining them does not significantly improve the manuscript. Hence, we removed this sentence from the manuscript.

**(8) Line 195-197: The simulated particle formation rates using these previous models and evaporation rates are orders of magnitude lower than the measured particle formation rates in urban Beijing. Why does this happen? Does this affect the robustness of your conclusion?**

**Response**: This is because the cluster evaporation rates in these models (Chen et al., 2012; Jen et al., 2014) were assumed high. We revised this sentence to emphasize that high evaporation rates are the main cause of the low particle formation rates: "However, due to these high evaporation rates, the simulated particle formation rates using these models are orders of magnitude lower than the measured particle formation rates in urban Beijing." These high evaporation rates were derived by matching the simulation results in these studies with their measurement results.

The difference between the kinetic model in this study and the previous models does not affect the robustness of the conclusions in this study. Instead, it indicates that the deficiencies of these models to simulate new particle formation rates in urban Beijing. The evaporation rate used in this study is consistent with quantum chemistry simulation results (Ortega et al., 2012; Myllys et al., 2019). The predicted dependencies of particle formation rate on the coagulation sink and amine concentration are consistent with measurements (Fig. 3).

The robustness of the conclusions was evaluated using the uncertainties in both measurements and simulations. Figure S5 shows that the simulation agrees with the measurements within the uncertainty range.

**(9) Line 251-253: You may want to directly give the saturation concentration of amines here.**

**Response**: Thanks. We added "(~5-20 ppt)" to this sentence.

**(10) Line 288-290: What are the main reasons for the much higher amine concentration in Shanghai than that in Beijing?**

**Response**: We do not know the exact reasons yet. Further efforts are needed to understand the sources of amine concentration in polluted environments.

**References**

Chen, M., Titcombe, M., Jiang, J., Jen, C., Kuang, C., Fischer, M. L., Eisele, F. L., Siepmann, J. I., Hanson, D. R., Zhao, J., and McMurry, P. H.: Acid-base chemical reaction model for nucleation rates in the polluted atmospheric boundary layer, Proceedings of the National Academy of Sciences of the United States of America, 109, 18713-18718, 10.1073/pnas.1210285109, 2012.

Deng, C., Fu, Y., Dada, L., Yan, C., Cai, R., Yang, D., Zhou, Y., Yin, R., Lu, Y., Li, X., Qiao, X., Fan, X., Nie, W., Kontkanen, J., Kangasluoma, J., Chu, B., Ding, A., Kerminen, V., Paasonen, P., Worsnop, R. D., Bianchi, F., Liu, Y., Zheng, J., Wang, L., Kulmala, M., and Jiang, J.: Seasonal characteristics of new particle formation and growth in urban Beijing, Environmental Science and Technology, 54, 8547–8557, 10.1021/acs.est.0c00808, 2020b.

Heinritzi, M., Simon, M., Steiner, G., Wagner, A. C., Kürten, A., Hansel, A., and Curtius, J.: Characterization of the mass-dependent transmission efficiency of a CIMS, Atmospheric Measurement Techniques, 9, 1449-1460, 10.5194/amt-9-1449-2016, 2016.

Jen, C. N., McMurry, P. H., and Hanson, D. R.: Stabilization of sulfuric acid dimers by ammonia, methylamine, dimethylamine, and trimethylamine, Journal of Geophysical Research: Atmospheres, 119, 7502-7514, 10.1002/2014jd021592, 2014.

Lehtipalo, K., Yan, C., Dada, L., Bianchi, F., Xiao, M., Wagner, R., Stolzenburg, D., Ahonen, L., Amorim, A., Baccarini, A., Bauer, P., Baumgartner, B., Bergen, A., Bernhammer, A.-K., Breitenlechner, M., Brilke, S., Buchholz, A., Mazon, S. B., Chen, D., Chen, X., Dias, A., Dommen, J., Draper, D. C., Duplissy, J., Enh, M., Finkenzeller, H., Fischer, L., Frege, C., Fuchs, C., Garmash, O., Gordon, H., Hakala, J., He, X., Heikkinen, L., Heinritzi, M., Helm, J. C., Hofbauer, V., Hoyle, C. R., Jokinen, T., Kangasluoma, J., Kerminen, V.-M., Kim, C., Kirkby, J., Kontkanen, J., Kürten, A., Lawler, M. J., Mai, H., Mathot, S., Mauldin, R. L., Molteni,

U., Nichman, L., Nie, W., Nieminen, T., Ojdanic, A., Onnela, A., Passananti, M., Petäjä, T., Piel, F., Pospisilova, V., Quéléver, L. L. J., Rissanen, M. P., Rose, C., Sarnela, N., Schallhart, S., Schuchmann, S., Sengupta, K., Simon, M., Sipilä, M., Tauber, C., Tomé, A., Tröstl, J., Väisänen, O., Vogel, A. L., Volkamer, R., Wagner, A. C., Wang, M., Weitz, L., Wimmer, D., Ye, P., Ylisirniö, A., Zha, Q., Carslaw, K. S., Curtius, J., Donahue, N. M., Flagan, R. C., Hansel, A., Riipinen, I., Virtanen, A., Winkler, P. M., Baltensperger, U., Kulmala, M., and Worsnop, D. R.: Multicomponent new particle formation from sulfuric acid, ammonia, and biogenic vapors, Science Advances, 4, eaau5363, 10.1126/sciadv.aau5363 2018.

Myllys, N., Kubečka, J., Besel, V., Alfaouri, D., Olenius, T., Smith, J. N., and Passananti, M.: Role of base strength, cluster structure and charge in sulfuric-acid-driven particle formation, Atmospheric Chemistry and Physics, 19, 9753-9768, 10.5194/acp-19-9753-2019, 2019.

Ortega, I. K., Kupiainen, O., Kurtén, T., Olenius, T., Wilkman, O., McGrath, M. J., Loukonen, V., and Vehkamäki, H.: From quantum chemical formation free energies to evaporation rates, Atmospheric Chemistry and Physics, 12, 225-235, 10.5194/acp-12-225-2012, 2012.

Riccobono, F., Schobesberger, S., Scott, C. E., Dommen, J., Ortega, I. K., Rondo, L., Almeida, J., Amorim, A., Bianchi, F., Breitenlechner, M., David, A., Downard, A., Dunne, E. M., Duplissy, J., Ehrhart, S., Flagan, R. C., Franchin, A., Hansel, A., Junninen, H., Kajos, M., Keskinen, H., Kupc, A., Kürten, A., Kvashin, A. N., Laaksonen, A., Lehtipalo, K., Makhmutov, V., Mathot, S., Nieminen, T., Onnela, A., Petäjä, T., Praplan, A. P., Santos, F. D., Schallhart, S., Seinfeld, J. H., Sipilä, M., Spracklen, D. V., Stozhkov, Y., Stratmann, F., Tomé, A., Tsagkogeorgas, G., Vaattovaara, P., Viisanen, Y., Vrtala, A., Wagner, P. E., Weingartner, E., Wex, H., Wimmer, D., Carslaw, K. S., Curtius, J., Donahue, N. M., Kirkby, J., Kulmala, M., Worsnop, D. R., and Baltensperger, U.: Oxidation products of biogenic emissions contribute to nucleation of atmospheric particles, Science, 344, 717-721, 10.1126/science.1243527, 2014.

Zapadinsky, E., Passananti, M., Myllys, N., Kurten, T., and Vehkamaki, H.: Modeling on Fragmentation of Clusters inside a Mass Spectrometer, J Phys Chem A, 123, 611-624, 10.1021/acs.jpca.8b10744, 2019.